# Partial rejuvenation of the spermatogonial stem cell niche after gender-affirming hormone therapy in trans women

**Emily Delgouffe[1]\*, Samuel Madureira Silva[1], Frédéric Chalmel[2], Wilfried Cools[3], Camille Raets[3], Kelly Tilleman[4], Guy T'Sjoen[5], Yoni Baert[1,6], Ellen Goossens[1]**

[1]Biology of the Testis (BITE) Laboratory, Genetics, Reproduction and Development (GRAD) Research Group, Vrije Universiteit Brussel, Brussels, Belgium; [2]Inserm, EHESP, Institut de Recherche en Santé, Environnement et Travail (IRSET), Université de Rennes, Rennes, France; [3]Core facility, Support for Quantitative and Qualitative Research (SQUARE), Vrije Universiteit Brussel, Brussels, Belgium; [4]Department for Reproductive Medicine, Ghent University Hospital, Ghent, Belgium; [5]Department of Endocrinology and Center for Sexology and Gender, Ghent University Hospital, Ghent, Belgium; [6]In Vitro Toxicology and Dermato-Cosmetology (IVTD), Vrije Universiteit Brussel, Brussels, Belgium

**\*For correspondence:** emily.delgouffe@vub.be

**Competing interest:** The authors declare that no competing interests exist.

**Sent for Review** 25 January 2024
**Preprint posted** 28 January 2024
**Reviewed preprint posted** 09 May 2024
**Reviewed preprint revised** 17 July 2024
**Version of Record published** 07 January 2025

## eLife assessment

This **important** study presents new knowledge of the spermatogonial stem cell (SSC) niche in trans women after gender-affirming hormone therapy (GAHT). The evidence supporting the claims is **convincing**. The work will be of interest to researchers and clinicians working in the field of reproductive medicine and andrology.

**Abstract** Although the impact of gender-affirming hormone therapy (GAHT) on spermatogenesis in trans women has already been studied, data on its precise effects on the testicular environment is poor. Therefore, this study aimed to characterize, through histological and transcriptomic analysis, the spermatogonial stem cell niche of 106 trans women who underwent standardized GAHT, comprising estrogens and cyproterone acetate. A partial dedifferentiation of Sertoli cells was observed, marked by the co-expression of androgen receptor and anti-Müllerian hormone which mirrors the situation in peripubertal boys. The Leydig cells also exhibited a distribution analogous to peripubertal tissue, accompanied by a reduced insulin-like factor 3 expression. Although most peritubular myoid cells expressed alpha-smooth muscle actin 2, the expression pattern was disturbed. Besides this, fibrosis was particularly evident in the tubular wall and the lumen was collapsing in most participants. A spermatogenic arrest was also observed in all participants. The transcriptomic profile of transgender tissue confirmed a loss of mature characteristics - a partial rejuvenation - of the spermatogonial stem cell niche and, in addition, detected inflammation processes occurring in the samples. The present study shows that GAHT changes the spermatogonial stem cell niche by partially rejuvenating the somatic cells and inducing fibrotic processes. These findings are important to further understand how estrogens and testosterone suppression affect the testis environment, and in the case of orchidectomized testes as medical waste material, their potential use in research.

## Introduction

Gender-affirming hormone therapy (GAHT) plays a crucial role in the care of many transgender individuals who seek alignment between their gender identity and the sex assigned at birth. Trans women may undergo GAHT which usually combines the administration of estrogens and anti-androgens to suppress endogenous testosterone production and to develop feminine secondary sexual characteristics (*Hembree et al., 2017*; *Coleman et al., 2022*). Following feminizing GAHT, trans women without any surgical contraindications can opt for a sex reassignment surgery (SRS) which typically consists of a bilateral orchidectomy, as well as a gender-affirming vulvoplasty or vaginoplasty (*van der Sluis et al., 2023*).

While various effects of feminizing GAHT on the physical transition have been extensively studied (*Schneider et al., 2017*; *Trans Primary Care, 2023*), its impact on the testicular microenvironment remains a subject of ongoing research. Previous studies mainly focused on the impact of GAHT on spermatogenesis, yielding contrasting outcomes, ranging from maturation arrest (*Lu and Steinberger, 1978*; *Sapino et al., 1987*; *Schulze, 1988*; *Payer et al., 2009*; *Leavy et al., 2017*; *Jindarak et al., 2018*; *Matoso et al., 2018*; *Schneider et al., 2019*; *Vereecke et al., 2020*; *de Nie et al., 2022a*) to full spermatogenesis (*Jiang et al., 2019*; *Alford et al., 2020*). Besides this, varying degrees of tubular hyalinization were found after GAHT (*Sapino et al., 1987*; *Schulze, 1988*; *Venizelos and Paradinas, 1988*; *Jindarak et al., 2018*; *Sinha et al., 2021*; *de Nie et al., 2022b*). Some studies also suggested dedifferentiation of Sertoli cells (*Schulze, 1988*) and/or dedifferentiation or loss of Leydig cells (*Rodriguez-Rigau et al., 1977*; *Sapino et al., 1987*; *Schulze, 1988*; *Venizelos and Paradinas, 1988*; *Kisman et al., 1990*; *Matoso et al., 2018*; *Cornejo et al., 2022*). These discrepancies may stem from the limited sample sizes in most studies and the differences in GAHT regimens, involving diverse hormone types, dosages, modes of administration or adherence to treatment (*Hembree et al., 2009*). Moreover, previous findings on the testicular cells mainly relied on histological outcomes and lacked the use of specific markers to accurately identify the condition of these cells.

Therefore, this study aims to thoroughly characterize the spermatogonial stem cell niche in a large cohort of trans women subjected to a standardized GAHT regimen. This research entails (immuno) histochemistry to examine the general condition of the tissue and the maturity/functionality of the somatic cells. Comparative differential gene expression and functional analysis are performed to compare testicular tissue of trans women with that of cisgender adults, as well as peri- and prepubertal boys.

**Table 1.** Participant characteristics.

| Parameter | Median (range) | n |
|---|---|---|
| Age at start GAHT (years) | 27.4 (16.0–69.0) | 106 |
| Duration of GAHT (years) | 1.8 (0.4–7.0) | 106 |
| Type of GAHT | | |
| Oral | 72.6% | 106 |
| Transdermal | 27.4% | 106 |
| Age at surgery (years) | 31.0 (18.1–70.1) | 106 |
| Time last visit to SRS (days) | 92 (1–764) | 104 |
| Serum hormone levels | | |
| LH (U/L) | 0.1 (0.09–9.6) | 104 |
| FSH (U/L) | 0.19 (0.09–12.0) | 104 |
| T (ng/dL) | 18.1 (0.1–499.1) | 104 |
| E2 (ng/L) | 66.0 (14.1–776.0) | 105 |
| AMH (µg/L) | 7.5 (0.03–143.0) | 68 |
| Inhibin B (pg/mL) | 59.5 (0.8–193.6) | 69 |

# Results

## Participant background

A summary of the participants' characteristics can be found in *Table 1*. At the time of orchidectomy, participants were 31.0 years of age (18.1–70.1) and had been administered GAHT for 1.8 years (0.4–7.0). All participants had reached adulthood at the time of SRS (Tanner stage G5). Most were treated with oral estrogen (72.6%) rather than transdermal estrogen (27.4%), and all participants received the anti-androgen cyproterone acetate (CPA). None of the participants had received prior treatment with puberty blockers. Serum hormone levels of the enrolled participants were recorded during the last pre-operative visit, 92 days (1–764) before SRS.

In this cohort, generally, the gonadotropins were sufficiently suppressed with serum luteinizing hormone (LH) levels of 0.10 U/L (0.09–9.6) and serum follicle-stimulating hormone (FSH) levels of 0.19 U/L (0.09–12.0). Both the Endocrine Society and the WPATH SOC 8 guidelines advise trans women to maintain serum testosterone (T) levels below 50 ng/dL (*Hembree et al., 2017*; *Coleman et al., 2022*). This threshold corresponds to the cisgender premenopausal female as well as the prepubertal male range (*Kulle et al., 2010*) and is met by 90.4% (94/104) of the included participants: 16.84 ng/dL (0.1–45.9). The remaining 10 participants (9.6%) exhibited higher T levels: 127.4 ng/dL (50.8–499.1). Notably, three of these individuals (2.9%) even showed T levels within the adult male range (>218 ng/dL) (*Travison et al., 2017*). In terms of serum estradiol (E2) levels, the threshold recommended by the Endocrine Society and WPATH SOC 8 guidelines (*Hembree et al., 2017*; *Coleman et al., 2022*) is set at 100–200 ng/L and corresponds to the cisgender premenopausal female range (*Kushnir et al., 2008*). Remarkably, 74.3% (78/105) of the participants showed E2 levels below this recommendation: 56.8 ng/L (14.1–94.7). Besides this, six participants (5.7%) exceeded this threshold: 410.5 ng/L (250.0–776.0). The remaining 21 participants (20.0%) displayed target E2 levels: 128.0 ng/L (102.0–188.0).

Regarding the serum anti-Müllerian hormone (AMH) levels, 56% (38/68) of the participants exhibited levels within the normal adult range: 5.2 µg/L (1.4–11.6). Noteworthy, 27 trans women (40%) surpassed the upper reference limit of 11.6 µg/L and fell within the male peripubertal range: 25.3 µg/L (11.8–143). Only three participants (4%) displayed low AMH levels: 0.42 µg/L (0.03–0.9). In contrast, inhibin B levels were predominantly low, as 77% (53/69) of trans women exhibited levels below the lower reference limit of 95 pg/mL: 55.3 pg/mL (0.8–93.7). Only 23% (16/69) of the trans women showed normal inhibin B levels: 140.7 pg/mL (95.9–193.6).

## Atypical histology and arrest of germ cell differentiation

Regarding the overall testicular histology, for each individual, one predominant phenotype could be observed for both lumen and hyalinization scoring. The prepubertal controls displayed an absent lumen, while peripubertal controls exhibited an absent to half-open lumen, and adult controls exhibited an open lumen. Out of the 106 trans women, 24 (23%) exhibited an open lumen, 51 (48%) a half-open lumen, and 31 (29%) an absent lumen (*Figure 1A*). Besides this, the controls exhibited minimal to no hyalinization. Some degree of tubular hyalinization could be detected in 78% of the trans women (83/106), of which 43 (41%) showed mild hyalinization, 31 (29%) moderate hyalinization, and 9 (8%) severe hyalinization (*Figure 1B*).

Regarding the status of spermatogenesis (*Figure 1C*), 79% (84/106) of the trans women did not exhibit spermatogenesis. They either showed spermatogonia only (67% MAGE[+]/BOLL[-]/CREM[-]/ACROSIN[-] [71/106]) or no germ cells at all (12% MAGE[-] [13/106]). The remaining 22 participants (21%) displayed maturation arrest at the level of secondary spermatocytes (8% only MAGE[+]/BOLL[+]/CREM[-]/ACROSIN[-] [8/106]), or round spermatids (13% MAGE[+]/BOLL[+]/CREM[+]/ACROSIN[-] [14/106]). Notably, no complete spermatogenesis (ACROSIN[+] cells) could be observed in this cohort.

For each of the outcomes of primary interest, which include 'tissue hyalinization', 'alpha smooth muscle actin 2 (ACTA2) pattern', 'progression of spermatogenesis', percentage of 'AMH[+]' tubules, and percentage of 'INSL3[+]' cells, a separate regression model was built to evaluate the relation with age at SRS, years of GAHT, and reproductive hormone levels (specifically LH, FSH, T, and E2). 'INSL3[+]' was continuously scaled and analyzed using a multiple linear regression model. 'AMH[+]' was continuously scaled but dichotomized into values equal to 100 or not, because too few values were below 100, and analyzed with a logistic regression model. Ordinally scaled variables 'tissue hyalinization', 'ACTA2 pattern', and 'progression of spermatogenesis' were examined through a cumulative logistic

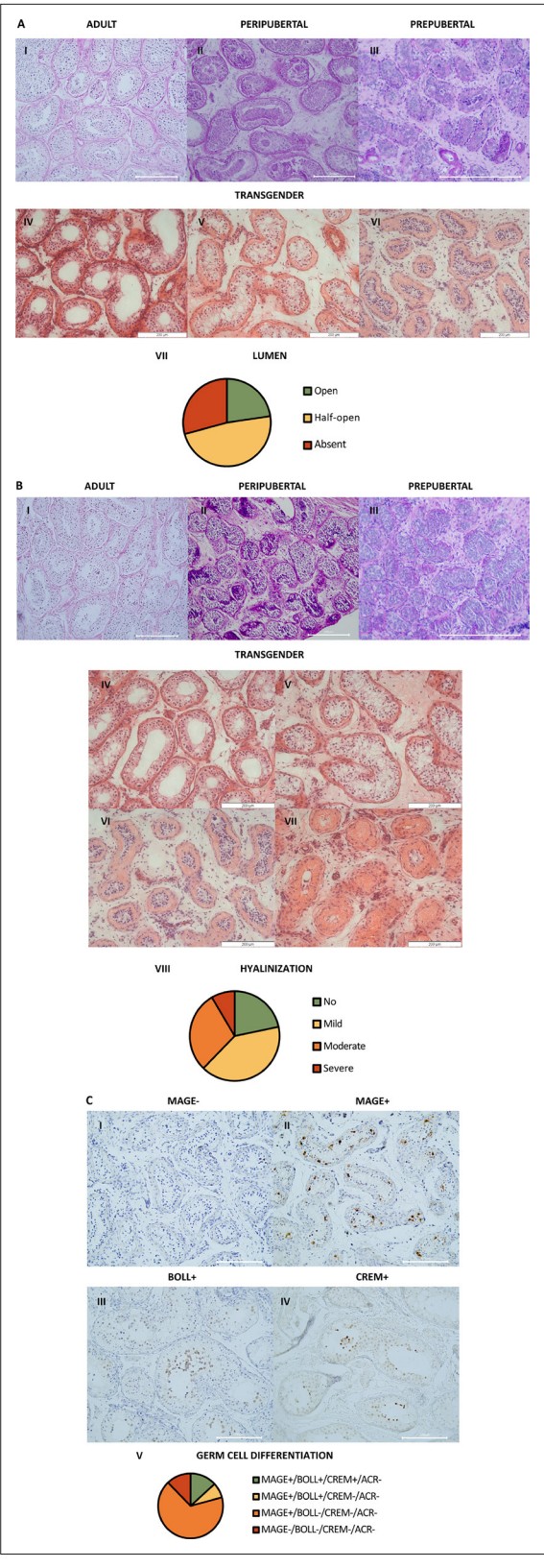

**Figure 1.** Atypical histology and arrest of germ cell differentiation in trans women. (**A**) shows representative pictures of the lumen in adult (**I**), peripubertal (**II**), and prepubertal (**III**) controls, and an open (**IV**), half-open (**V**), or absent (**VI**) lumen in trans women. Graph **VII** shows the distribution of trans women over the three lumen categories. For each participant, the category was determined by the predominant pattern. (**B**) illustrates the

*Figure 1 continued on next page*

*Figure 1 continued*

hyalinization status in adult (**I**), peripubertal (**II**), and prepubertal (**III**) controls, and the hyalinization stages that could be observed in the trans women: no (**IV**), mild (**V**), moderate (**VI**), or severe (**VII**) hyalinization. Graph **VIII** shows the distribution of trans women over the hyalinization stages. For each participant, the category was determined by the predominant pattern. (**C**) demonstrates the different stages of germ cell differentiation that were present in trans women, namely MAGE$^-$/BOLL$^-$/CREM$^-$/ACROSIN$^-$ (**I**), MAGE$^+$/BOLL$^-$/CREM$^-$/ACROSIN$^-$ (**II**), MAGE$^+$/BOLL$^+$/CREM$^-$/ACROSIN$^-$ (**III**), and MAGE$^+$/BOLL$^+$/CREM$^+$/ACROSIN$^-$ (**IV**). No ACROSIN$^+$ cells were detected. Graph **V** illustrates the distribution of trans women over the different stages of germ cell differentiation. Scale bars represent 200µm. MAGE: melanoma-associated antigen A4, BOLL: boule homolog RNA-binding protein, CREM: cAMP-responsive element modulator.

regression approach. The selection of variables for inclusion in the regression models was based on the Akaike information criterion.

The cumulative logistic regression model for 'progression of spermatogenesis' retained only the predictors 'age at SRS' and 'LH'. 'Age at SRS' estimated at –0.081 suggests that as participants are older, they are more likely to show MAGE$^-$/BOLL$^-$/CREM$^-$/ACROSIN$^-$ or MAGE$^+$/BOLL$^-$/CREM$^-$/ACROSIN$^-$ phenotypes. 'LH' estimated at 1.535 suggests that higher LH values are indicative for BOLL$^+$, and even more for CREM$^+$.

## Peritubular myoid cells show a distinct ACTA2 pattern

Although ACTA2 expression is absent in the PTMCs of prepubertal boys, during puberty, it appears in certain seminiferous tubules, resulting in an 'interrupted' pattern (*Figure 2*). At adulthood, all seminiferous tubules are surrounded by an 'intact', dense layer of ACTA2-expressing PTMCs. Interestingly, in most transgender participants (49% [51/104]), a distinct, thicker ACTA2 pattern could be observed, in which the PTMCs appear 'disconnected' from each other (*Figure 2VIII*). The other transgenders either showed an 'intact' (29% [30/104]), 'interrupted' (16% [17/104]), or an 'absent' (6% [6/104]) ACTA2 pattern. It was further investigated whether this thicker 'disconnected' ACTA2 pattern might be linked to the age of the participant. While a slightly thicker ACTA2$^+$ layer could be observed in aged controls, the PTMCs did not exhibit the 'disconnected' appearance seen in transgender participants.

The cumulative logistic regression model for 'ACTA2 pattern' retained 'age at SRS' as well, together with 'log-T'. 'Age at SRS' estimated at –0.057 suggests that as participants are older, they are more likely to show an interrupted ACTA2 pattern, and sometimes even an absent pattern. 'Log-T' estimated at 0.677 suggests that higher log-T values are indicative of an intact ACTA2 pattern, and slightly less disconnected patterns. The analysis also pointed out that 'interrupted' and 'absent' ACTA2 patterns did not occur much.

## Partial Sertoli cell dedifferentiation

As presented in *Figure 3A*, the Sertoli cells of prepubertal boys express both SRY-box transcription factor 9 (SOX9) and AMH, but not androgen receptor (AR). However, in peripubertal boys, the AMH signal starts to vanish while the AR signal emerges. By the time adulthood is reached, Sertoli cells are positive for SOX9 and AR and no longer express AMH. For the evaluation of the AMH signal, expression patterns could be established based on the control tissues. Briefly, the 'prepubertal' pattern showed a strong AMH signal in all seminiferous tubules, which always overlapped with the SOX9 signal. In the 'peripubertal' pattern, there was a moderate AMH signal overlapping with the SOX9 signal within the seminiferous tubules, but also a stronger AMH signal in the tubular wall that did not overlap with the SOX9 signal. The 'adult' pattern only showed an AMH signal within the tubular wall which never overlapped with the SOX9 signal. Therefore, the 'adult' pattern was considered as AMH$^-$. Interestingly, it has been shown that AMH receptor type 2 starts to be expressed in peritubular mesenchymal cells within the tubular walls during puberty and it remains so throughout adulthood (*Sansone et al., 2020*). AMH bound to this receptor may help explain the observed AMH signal in the tubular wall of peripubertal and adult controls. The predominant AMH pattern among transgender individuals was the prepubertal pattern (75% [79/106]), followed by the peripubertal pattern (18% [19/106]) and the adult pattern (7% [8/106]) (*Figure 3B*).

*Figure 3C* shows the presence of 'mature' (SOX9$^+$/AR$^+$/AMH$^-$), 'semi-mature' (SOX9$^+$/AR$^+$/AMH$^+$), and 'immature' (SOX9$^+$/AR$^-$/AMH$^+$) tubules in trans women. The seminiferous tubules of nearly all

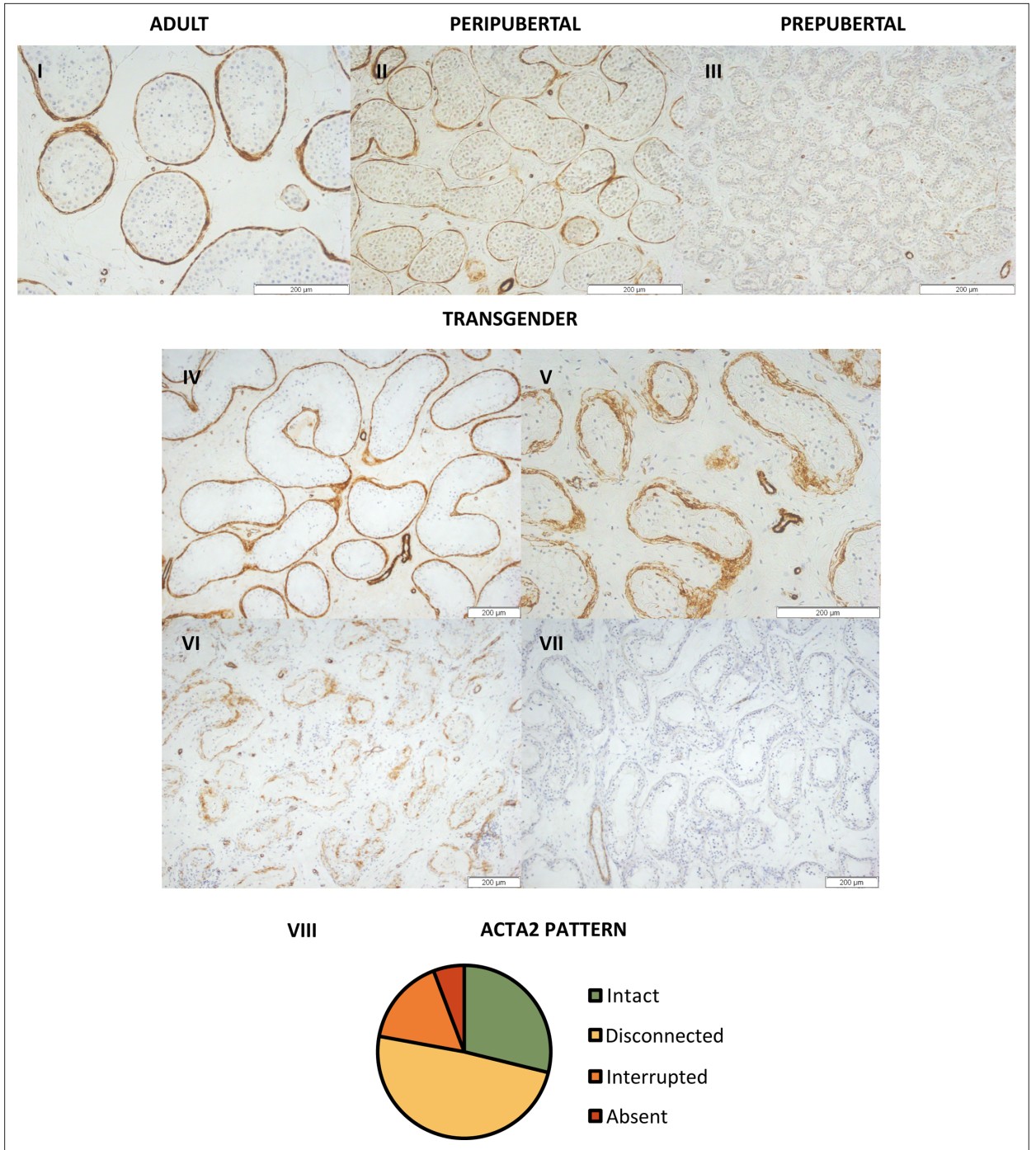

**Figure 2.** Distinct ACTA2 expression patterns in the peritubular myoid cells of trans women. Representative images of the ACTA2 expression patterns in adult (**I**), peripubertal (**II**), and prepubertal control (**III**), together with the 'intact' (**IV**), 'disconnected' (**V**), 'interrupted' (**VI**), and 'absent' (**VII**) patterns seen in trans women. Panel **VIII** illustrates the distribution of trans women across the four ACTA2 patterns. For each participant, the ACTA2 category was determined by the predominant pattern. Scale bars represent 200 µm. ACTA2: alpha-smooth muscle actin 2.

participants (93% [99/106]) co-express SOX9, AR, and AMH. In most cases, these participants showed solely semi-mature tubules (85% [90/106]), but some trans women displayed a mix of both mature and semi-mature tubules (3% [3/106]) or semi-mature and immature tubules (6% [6/106]). Among these mixed phenotypes, the presence of semi-mature tubules accounted for 20.4% (0.4–48.4) and 91.1% (81.7–99.3) of the total tubule count, respectively. Notably, only seven participants (7%) exhibited exclusively mature tubules and none of the participants had only immature tubules. Furthermore, the

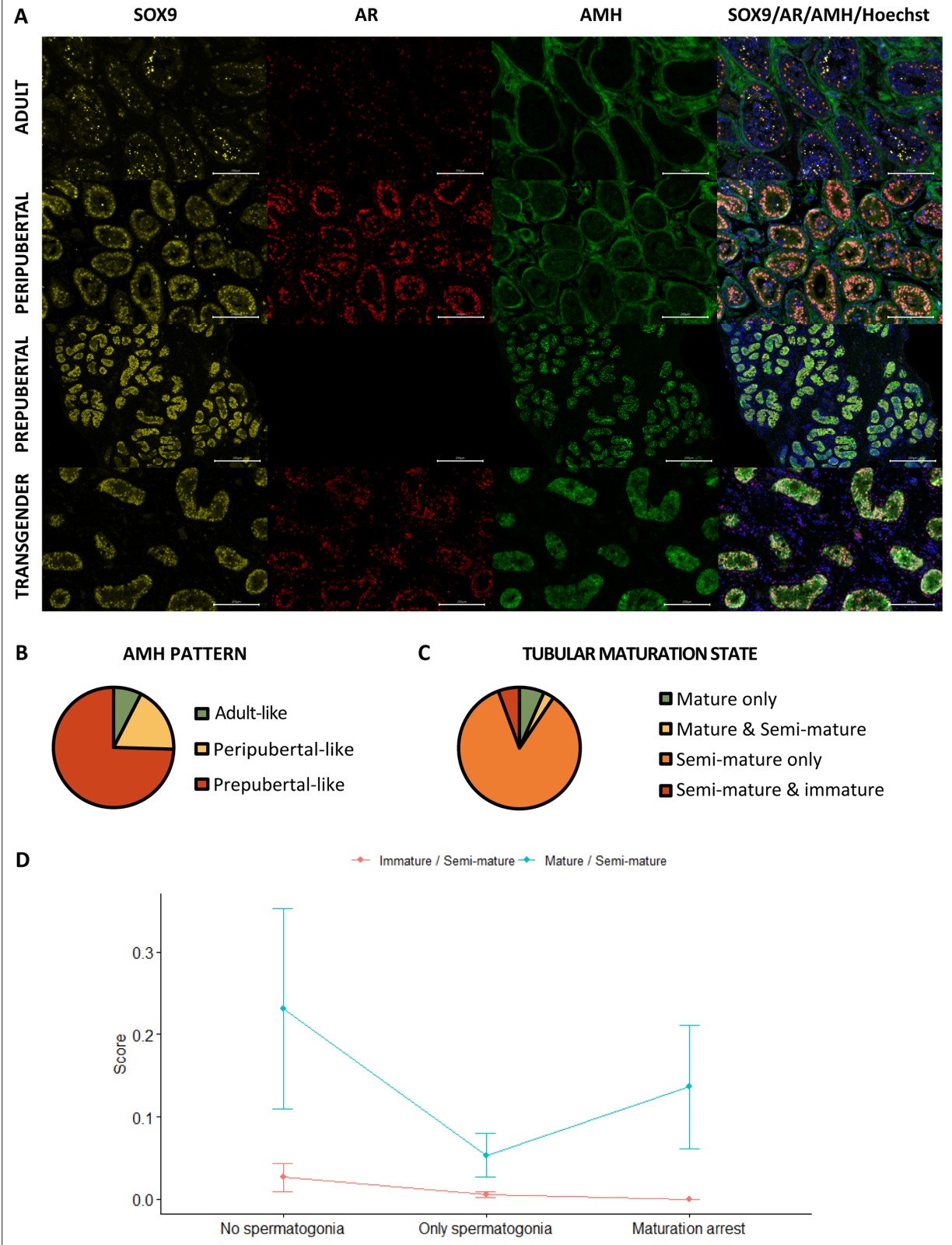

**Figure 3.** Partial Sertoli cell dedifferentiation in trans women. (**A**) illustrates the presence of SOX9 (general marker), AR (mature marker), and AMH (immature marker) within the seminiferous tubules of adult, peripubertal, and prepubertal control tissues, as well as in transgender tissue. Graph **B** shows the presence of the different AMH expression patterns in trans women. Additionally, graph **C** summarizes the percentage of trans women with only mature tubules, both mature and semi-mature tubules, only semi-mature tubules, and a combination of semi-mature and immature tubules. Graph

*Figure 3 continued on next page*

*Figure 3 continued*

**C** shows the presence of the different AMH expression patterns in trans women. Graph **D** illustrates the variation in Sertoli cell maturation across the different spermatogenic conditions. Scale bars represent 200 µm. Data are shown as mean ± standard error. SOX9: SRY-box transcription factor 9, AR: androgen receptor, and AMH: anti-Müllerian hormone.

logistic regression model for 'AMH$^+$' retained only 'log-T' as a predictor. 'Log-T' estimated at –0.813 suggests a decrease in the probability of having 100% AMH$^+$ tubules as T levels increase.

Additional analyses were performed to evaluate differences between the spermatogenesis conditions ('No spermatogonia': MAGE-, 'Only spermatogonia': MAGE+, and 'Partial differentiation': BOLL+ and CREM+). The percentages of tubules with mature/semi-immature/immature Sertoli cells for each of the three conditions were often (near) 0 or 100, for which a tangent hyperbolic transformation was performed on the ratios of percentages, scaling the ratios within the interval –1 to 1. Note that the three percentages of the three spermatogenesis conditions are mutually dependent, each one depending on the others, which we dealt with by recombining the three conditions into two ratios or odds $\left( \frac{\%\text{Immature}}{\%\text{Semi-mature}} \text{ and } \frac{\%\text{Mature}}{\%\text{Semi-mature}} \right)$. These percentages, and thus the transformed ratios, were observed with two types of ratios for which a repeated measures ANOVA was used on the ranks of the transformed scores, to evaluate the differences between conditions. The significant interaction effect suggests that differences between conditions are not the same for the two types of odds (relative percentage of immature or relative percentage of mature) (p=8.131e-05). While there is also evidence for a significant difference between the two types, because of the interaction this difference can only be interpreted with caution. The transformed ratio mature/semi-mature tubules does seem higher than the ratio immature/semi-mature tubules (p=2.127e-12) (*Figure 3D*). Furthermore, as the condition of the spermatogenesis improves, a decreasing trend is noted for the ratio immature/semi-immature tubules. For the ratio mature/semi-immature tubules, the score is the highest for the 'No spermatogonia' condition, followed by the 'Partial differentiation' condition.

A correlation analysis was performed to assess whether the abundant presence of AMH signal in the tissue is also reflected in the serum AMH levels and to explore the possible connections between serum AMH and other variables of interest ('age at SRS', 'years of GAHT', 'serum LH', 'serum FSH', 'serum T', 'serum E2', 'serum inhibin B', 'tissue hyalinization', 'ACTA2 pattern', 'progression of spermatogenesis', and 'percentage of INSL3$^+$ cells'). Serum AMH values from 66 participants could be included in this analysis. Significant correlations were found for 'serum AMH' and the 'percentage of AMH$^+$ tubules' (r=0.330), 'tissue hyalinization' (r=−0.289), and 'progression of spermatogenesis' (r=−0.278).

## Reduced Leydig cell maturity and functionality

In prepubertal boys, neither cytochrome P45011 (CYP11a1) nor insulin-like factor 3 (INSL3) is expressed within the Leydig cells. From early puberty onward, both markers are distinctly expressed (*Figure 4A*). The functional maturity rate of the Leydig cells (CYP11a1$^+$/INSL3$^+$) in the adult controls ranged from 72% to 95%. More than half of the transgender participants showed >60% functionally mature Leydig cells (55% [57/103]), with the majority scoring >80% (31% [31/103]). In contrast, the remaining 46 participants displayed 41–60% (17% [17/103]) or 21–40% functionally mature Leydig cells (17% [17/103]), and 12 participants (12%) even scored below 20% (*Figure 4B*). Another noteworthy observation in transgender individuals was the tendency of Leydig cells to form smaller clusters in comparison to the adult controls, resembling what was seen in peripubertal controls. Additionally, in a few isolated cases, a single cell could be observed that was CYP11a1$^-$ but INSL3$^+$.

The linear regression model for 'INSL3$^+$' retained 'age at SRS' and the newly defined 'combination of FSH and LH'. 'Age at SRS' estimated at –0.595 suggests that as participants are older the INSL3$^+$ score decreases. The participants with FSH<0.3 appear to show fewer INSL3$^+$ cells than participants with FSH>0.3 and LH>0.1, estimated to differ –19.460. The remaining category (FSH>0.3 and LH<0.1) does not differ from the other two.

The percentages of the Leydig cells for each of the three conditions are also often (near) 0 or 100, for which again a hyperbolic tangent transformation was performed on the ratios of percentages. The transformed ratio of percentages $\left( \frac{\%\text{ functional cells}}{\%\text{afunctional cells}} \right)$ was analyzed with a Kruskal-Wallis test to evaluate differences between conditions. A significant effect for the conditions (p=0.002), as presented in

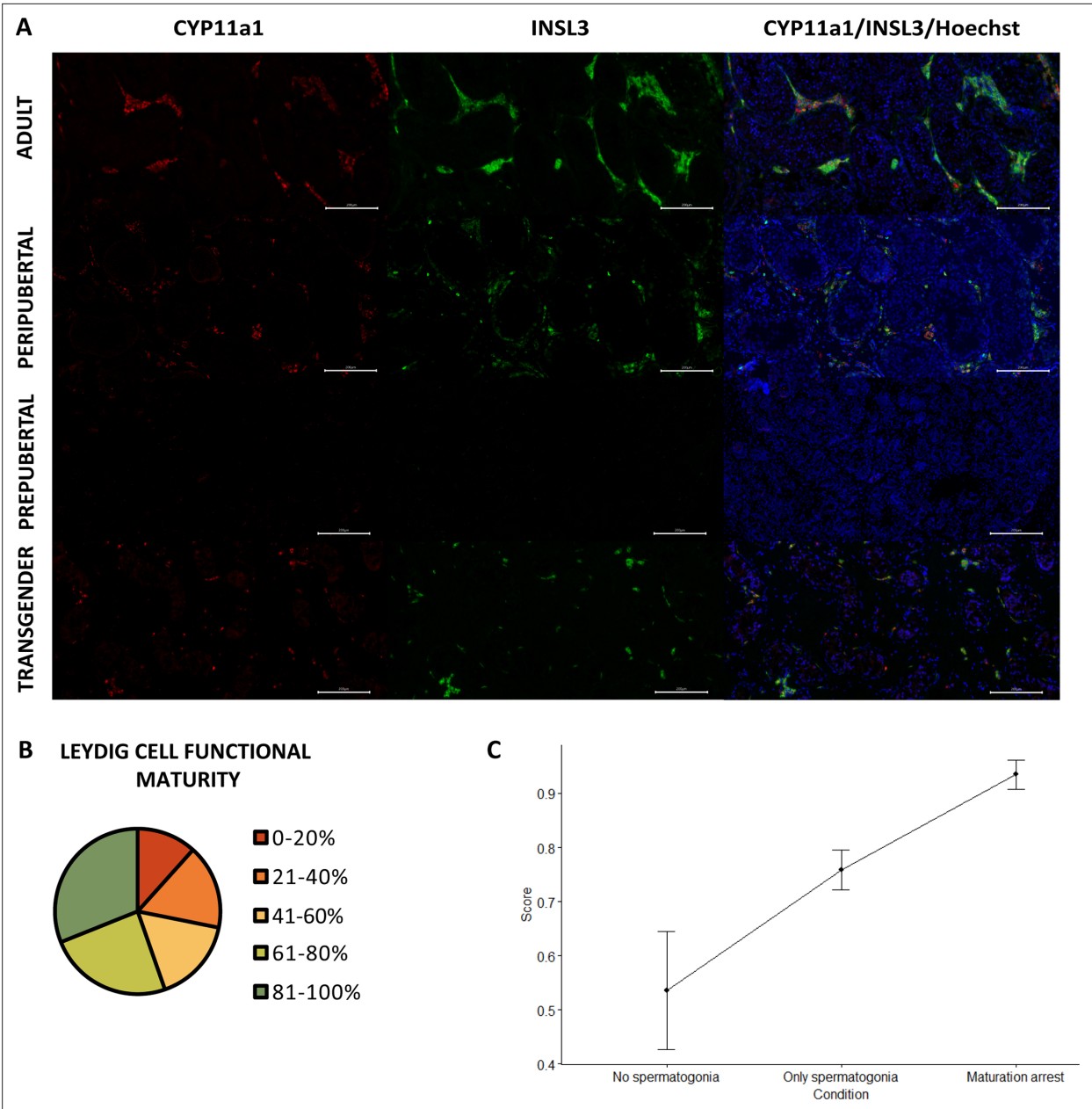

**Figure 4.** Reduced Leydig cell maturity and functionality in trans women. (**A**) illustrates the presence of CYP11a1 (functional marker) and INSL3 (mature marker) within adult, peripubertal, and prepubertal control tissues, as well as in transgender tissue. Graph **B** categorizes the Leydig cell functional maturity. Graph **C** shows the difference in functional maturity between the conditions of spermatogenesis. Scale bars represent 200 μm. Data are shown as mean ± standard error. CYP11a1: cytochrome P45011 and INSL3: insulin-like factor 3.

*Figure 4C*, resulted in a further exploration with the Wilcoxon rank sum test for the different pairwise combinations, with Bonferroni correction. Significant differences could be observed between the 'No spermatogonia' and 'Partial differentiation' groups (p=0.006) and between the 'Only spermatogonia' and 'Partial differentiation' groups (p=0.019). No significant differences were observed between the 'No spermatogonia' and 'Only spermatogonia' groups (p=0.173). It was noted that the score in the 'Partial differentiation' group was higher compared to the 'No/Only spermatogonia' groups, suggesting that there are relatively more functionally mature Leydig cells in the 'Partial differentiation' than in the 'No/Only spermatogonia' groups.

## Transcriptomic resemblance between transgender and immature testicular tissue

To compare the different types of testicular tissue at gene expression level, testicular tissue transcriptomes were obtained from trans women (n=6), and adult (n=5), peripubertal (n=3), and prepubertal (n=3) controls. Following data normalization, the 17 transcriptomes were projected on a two-dimensional PCA-based space. The first component explaining approximately 90% of the variation (x-axis) differentiated samples according to progression of spermatogenesis and tissue maturity (*Figure 5A*), locating the transgender transcriptomes next to the prepubertal ones. The peripubertal transcriptomes are also segregated according to the progression of spermatogenesis and tissue maturity. The least advanced (Peripubertal 1) is positioned next to prepubertal and transgender transcriptomes, while the most advanced (Peripubertal 3) is located next to the adult control transcriptomes and the intermediately advanced (Peripubertal 2) in the middle of both transcriptome groups.

When comparing the four different types of tissue, 11,661 differentially expressed genes (DEGs) were found (*Figure 5B*). Most DEGs were found between prepubertal and adult control (9944 DEGs), and adult control and transgender (8563 DEGs). On the other hand, the least DEGs were found between prepubertal and transgender tissue (1676 DEGs), highlighting the similitude between these two tissues also at a transcriptomic level. The DEGs were subsequently clustered into five expression patterns (P1-P5) to understand how the different testicular tissue types relate to each other (*Figure 5C*). Functional analysis revealed 1689 enriched GO terms to be significantly associated with the five expression patterns (*Supplementary file 7* provides all enriched GO terms). The top and more revealing GO terms for each pattern were selected and displayed next to the five expression patterns in *Figure 5C* with their corresponding number of associated genes and p-value. Pattern P1 is characterized by biological processes related to tissue morphogenesis and development. As expected, prepubertal tissue shows the highest gene expression for this pattern, as the tissue is in development. Interestingly, transgender tissue shows a gene expression intensity similar to that in peripubertal tissue (particularly donor Peripubertal 1) and between adult and prepubertal tissue. For pattern P2, mainly cellular components were significantly enriched in the GO terms analysis, which means that this pattern describes the location where activities and actions of these gene products are executed. Contrary to pattern P1, P2 can be related to a more stable tissue environment, where most activities happen outside the cells providing structural support, and biochemical or biomechanical cues. Consequently, for P2, adult control tissue shows the highest expression, and, again, transgender tissue has a similar expression to donor Peripubertal 1. Pattern P3 represents meiosis and sperm production. As expected, since spermatogenesis is repressed in trans women, transgender tissue showed a low expression of genes in this pattern similar to prepubertal tissue. P4 and P5 are the patterns that set transgender tissue apart from all other tissue types. P4 is associated with cell components (chromosome and spindle) important in both mitotic and meiotic processes. In prepubertal tissue, proliferation and tissue morphogenesis (mitosis) are present, while spermatogenic activity (meiosis) is part of adult tissue. Thus, as expected, less gene expression for P4 was observed in transgender tissue since it is structurally adult (no mitosis) but does not host spermatogenesis (no meiosis). P5 is a gene expression pattern specific to transgender tissue. This pattern includes several biological processes pointing to active inflammation and fibrosis described by the GO terms 'cell population proliferation' (119 genes, adjusted p-value<5.90e-15), which is not necessarily related to the mitotic cell cycle, but rather an increase in the number of cells in a certain space over a period (due to cell infiltration), 'cell activation' (96, 1.69e-13) or 'leukocyte activation' (83, 5.19e-11), 'cell migration' (95, 1.59e-11), 'cell death' (107, 1.38e-07), 'regulation of cell differentiation' (83, 5.46e-07) if associated with macrophages, e.g., 'response to stimulus' (348, 2.39e-16), 'cellular response to cytokine stimulus' (72, 3.05e-10), 'response to growth factor' (49, 7.02e-07), 'regulation of multicellular organismal process' (167, 4.21e-22), and 'extracellular matrix organization' (44, 8.84e-13). In addition, commonly associated with inflammation and fibrosis, tumorigenesis-related processes suggested by the GO terms 'angiogenesis' (56, 4.43e-14) and 'MAPK cascade' (53, 9.85e-06) are also part of P5. Highlighting these findings, trans woman 6 shows the highest expression for this pattern, which is also the most fibrotic tissue out of the six transgender tissues used for the gene expression analysis (pictures not shown). Also noteworthy, is the GO term 'response to estrogen' (8, 1.55e-2) in P5, suggesting the particularity of this expression pattern to transgender tissue. Thus, the biological processes in P5 show in what ways transgender tissue is divergently transforming from the other tissue types. Taken together, these findings suggest

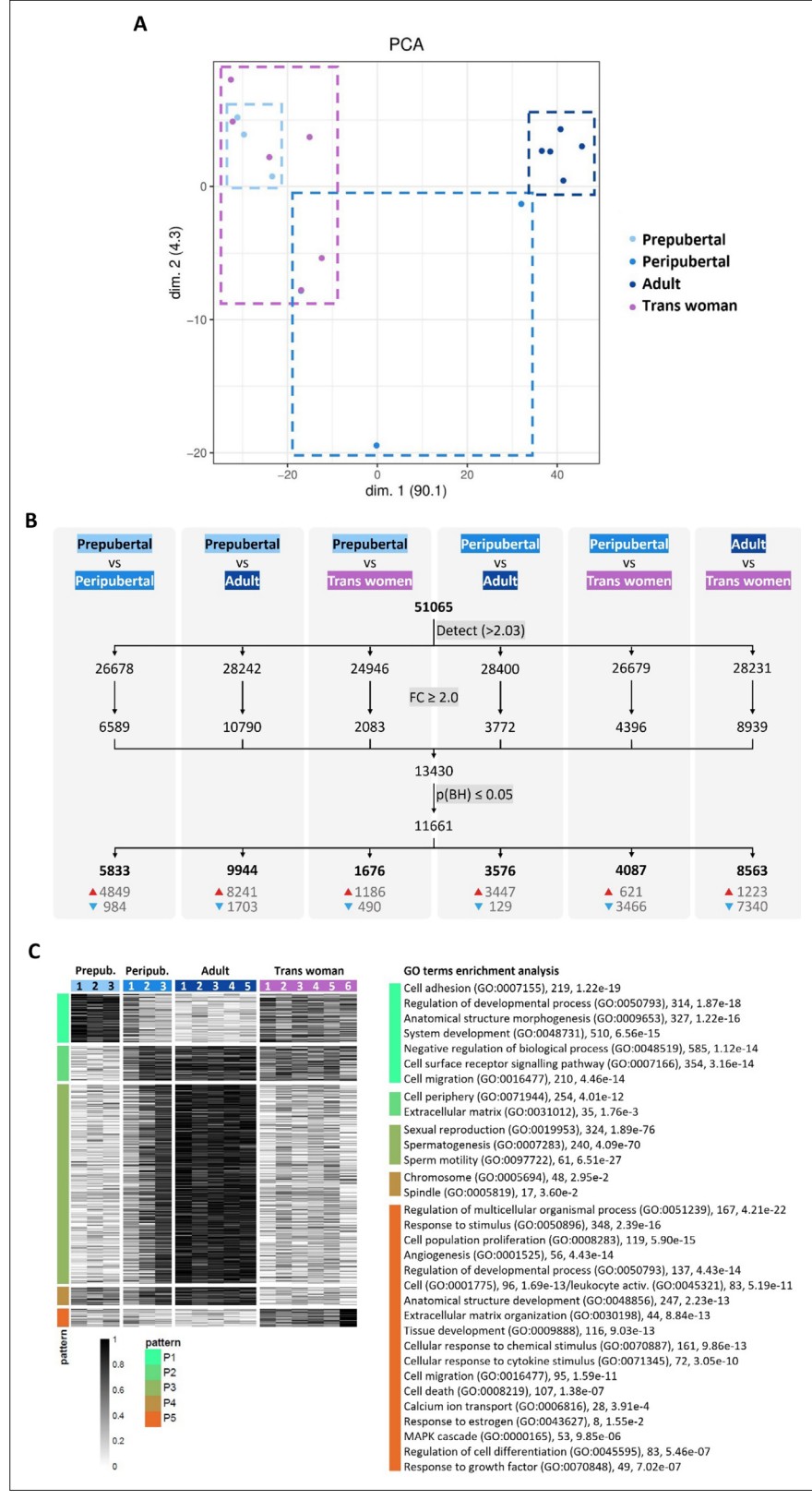

**Figure 5.** Transcriptome profile of control and transgender testicular tissue. Projection on a two-dimensional PCA-based space of preprocessed sample data (**A**). The first dimension (dim. 1) represents around 90% of the variance in the data, segregating samples into two distinct groups: trans women/prepubertal and adult. Peripubertal samples were distributed across dim. 1. For each comparison (**B**), the number of genes above the background

*Figure 5 continued on next page*

*Figure 5 continued*
expression cutoff (2.032) is given (a total of 51,065 were analyzed). The threshold for identifying differentially expressed genes (DEGs) was set to ≥2.0 fold change (FC) and significance was determined using an adjusted p-value of ≤0.05 (**B**). The numbers at the bottom of the columns show upregulated (red arrow) and downregulated (blue arrow) DEGs. Heatmap representation of DEGs between the different types of testicular tissue analyzed (**C**). In total, 11,661 DEGs were detected and grouped into five expression patterns (P1–P5) highlighting similarities and differences between the tissue groups. Each row represents a gene and each column a different donor. On the right side, GO terms enrichment analysis reveals biological processes (and some cellular components) significantly associated with each expression pattern. Next to each GO term, the number of associated genes and the corresponding p-value are provided.

that GAHT can significantly change the adult testis transcriptome to a less mature one, as well as initiate other biological processes that are not normally activated in the testis transcriptome.

## Discussion

As many individuals with gender dysphoria seek medical treatment, the need to comprehend the impact of hormone therapy on their bodies, and more particularly on their fertility increases. Despite existing research on the effect of GAHT on spermatogenesis in trans women, the exact influence on the testicular environment remains inadequately explored. This study aimed to fill this knowledge gap by thoroughly examining the spermatogonial stem cell niche of 106 trans women subjected to a standardized GAHT regimen, consisting of estrogens and CPA.

Upon analyzing the testicular histology, it became evident that most trans women either showed a half-open or absent lumen, aligning with the observations in the peri- and prepubertal controls, respectively. Only 23% of the trans women displayed an open lumen, resembling the normal status in adults. Moreover, unlike the controls, a significant portion of trans women displayed mild (41%), moderate (29%), or severe (8%) tubular hyalinization. Earlier studies already brought up varying degrees of hyalinization, basal membrane thickening, and seminiferous tubule atrophy as a result of GATH (*Sapino et al., 1987*; *Schulze, 1988*; *Venizelos and Paradinas, 1988*; *Schneider et al., 2019*). More recent work on 50 trans women, treated with varying types of GAHT, demonstrated a thickening of the tubular wall in 82% of the participants (*Matoso et al., 2018*). Similarly, in another examination of 136 adult trans women undergoing estrogen and CPA treatment, a majority exhibited tubular hyalinization, ranging from 76% to 92%, depending on the cessation of the GAHT for 4 weeks, or its continuation before SRS, respectively (*de Nie et al., 2022b*). Additionally, most of these participants exhibited an absent or half-open lumen, regardless of whether the GAHT was discontinued. Conversely, two other recent studies, one involving 173 trans women undergoing mixed GAHT (*Jindarak et al., 2018*), and another one including 85 trans women treated with estrogen and spironolactone (*Sinha et al., 2021*), only reported hyalinization in 6–28% of the participants, respectively. These variations in results could be attributed to differences in study methodologies as the latter two studies focused exclusively on severe cases of hyalinization, while both our study and the one conducted by De Nie and colleagues considered a broader spectrum, encompassing milder forms of hyalinization.

As we reported earlier, most trans women in this cohort lacked ongoing spermatogenesis, primarily showing spermatogonia-only patterns (*Vereecke et al., 2020*). This previous report mentioned a positive correlation between the number of spermatogonia and the serum T, LH, and inhibin B levels, and a negative correlation with the serum AMH levels and the age at the time of surgery. In the current study, a negative correlation could be observed between the serum AMH levels and the degree of tubular hyalinization and spermatogenesis. Moreover, the regression analysis indicated that higher LH values were associated with differentiation up to secondary spermatocytes and, even more, with differentiation up to round spermatids, while an older age at the time of surgery was indicative of a spermatogonia-only pattern or the complete absence of germ cells.

Examining the PTMCs, a major component of the tubular wall, was crucial to delve deeper into the abnormalities observed in the tubular wall of trans women. The PTMCs provide structural support to the seminiferous tubules, contribute to their contractile function, and exhibit a smooth muscle phenotype characterized by intensive expression of ACTA2, distinguishing them from fibroblasts (*Volkmann et al., 2011*; *Mayerhofer, 2013*). Our results indicated that nearly half of the included trans women

showed a 'disconnected' ACTA2 pattern. The occurrence of this phenotype might be linked to excessive deposition of extracellular matrix, causing spacing between the PTMCs and a more intensive thickening of the tubular wall. Moreover, 16% of the trans women displayed an 'interrupted' pattern, while 6% presented an 'absent' pattern, reminiscent of the patterns of peripubertal and prepubertal boys, respectively. However, by looking at the morphology of the tissue, the latter two patterns are more likely a reflection of the hyalinization within the tubular wall, causing a loss of the PTMCs rather than their rejuvenation. Noteworthy, our regression analysis revealed that as trans women are older at the time of surgery, they are more likely to have an 'interrupted' or an 'absent' pattern. Nevertheless, these disturbed patterns do not seem to be age-related, as older adult controls only exhibited a slightly thicker ACTA2 layer, without a 'disconnected' or 'interrupted' appearance. Apart from this, ACTA2 expression is stimulated by androgens (*Schlatt et al., 1993*) which are suppressed in trans women undergoing GAHT. This may result in a loss of ACTA2 expression, causing a shift in the PTMC phenotype as smooth muscle cells are known to do so in response to changes in the local environment (*Owens et al., 2004*). Adult Klinefelter syndrome participants, for example, also typically possess low T levels and exhibit an 'interrupted' or 'absent' ACTA2 pattern (*Van Saen et al., 2020*). Interestingly, the regression analysis pointed out that higher serum T values were indicative of 'intact' ACTA2 patterns, and to a somewhat lesser extent of 'disconnected' ACTA2 patterns.

Since it was previously suggested that the Sertoli cells of trans women might transform into immature cells (*Schulze, 1988*), their maturation state was of particular interest. Almost all trans women in this cohort exhibited only semi-mature Sertoli cells, expressing both the mature marker AR and the immature marker AMH, resembling the peripubertal controls. A minority of the trans women in this cohort showed mixed phenotypes, in which the presence of semi-mature tubules accounted for a significant portion of the total tubule count. Only a few trans women exclusively exhibited mature Sertoli cells, and none showed solely immature tubules. As T is essential for maintaining the differentiated phenotype and function of Sertoli cells (*Gholami et al., 2023*), these findings suggest that the disrupted steroid hormone levels after GAHT could cause a partial dedifferentiation of the Sertoli cells (*Nistal et al., 2013*). Upon comparing Sertoli cell maturity among the different stages of spermatogenesis, it became apparent that although the group without spermatogonia exhibited the highest ratio of mature/semi-mature tubules, those with only spermatogonia and partial differentiation followed an increasing trend. Moreover, the ratio of immature/semi-mature tubules decreased as participants displayed more advanced stages of germ cell differentiation. These observations could possibly be attributed to the older age of participants within the group without spermatogonia, with an age of 46.3±12.7 years old compared to 34.3±13.0 years old in the spermatogonia-only group and 27.9±8.7 years old in the group with partial differentiation. A less pronounced age difference was also noted between the only spermatogonia and partial differentiation groups. Such an age-related variation aligns with earlier findings on this cohort, indicating that suppressing spermatogenesis might be more challenging in younger individuals (*Vereecke et al., 2020*). The age-related differences in drug sensitivity, particularly the higher metabolization in younger people (*Cherry and Morton, 1989*), or the use of transdermal estrogens in older participants could contribute to the observed trend. Nevertheless, the age at SRS did not emerge as a predictor for the number of AMH+ tubules, whereas elevated serum T levels were contraindicative for having 100% AMH+ tubules. Previous research also indicated a correlation between intratesticular AMH levels and the duration of GAHT (*Schneider et al., 2021*). Our study did not find a correlation between serum AMH levels and GAHT duration, but there was a significant positive correlation between serum AMH and the percentage of AMH+ tubules.

The significant role of Leydig cells in male masculinization prompts intriguing questions about their status in trans women undergoing GAHT. Earlier work already mentioned a reduction in their number (*Rodriguez-Rigau et al., 1977*; *Sapino et al., 1987*; *Venizelos and Paradinas, 1988*; *Kisman et al., 1990*; *Payer et al., 2009*; *Sinha et al., 2021*; *Cornejo et al., 2022*), their regression to an immature phenotype (*Matoso et al., 2018*), or their dedifferentiation into fibroblast-like cells (*Schulze, 1988*). Something that immediately stood out in the present analysis was the fact that, contrary to adult controls, trans women exhibited very small Leydig cell clusters, reminiscent of what was observed in peripubertal controls. Interestingly, estrogens promote Leydig cell engulfment by macrophages by stimulating Leydig cells to produce growth arrest-specific 6, which mediates phagocytosis of apoptotic cells by bridging cells with surface-exposed phosphatidylserine to macrophage receptors (*Yu et al., 2014*). INSL3 was used to further assess the Leydig cells, as it is exclusively produced by Leydig

cells and is only indirectly regulated by the hypothalamic-pituitary-gonadal axis, making it a reliable marker to assess the Leydig cell population size and Leydig cell maturity (*Ivell et al., 2013*; *Anand-Ivell et al., 2022*). Nevertheless, it should be considered that excessive E2 concentrations might suppress INSL3 expression (*Laguë and Tremblay, 2009*; *Sansone et al., 2019*). Within this cohort, the functional maturity of the Leydig cells varied widely, ranging from 2% to 97%. Only in 47% of the trans women, the functional maturity of the Leydig cells exceeded 70%, which was similar to the adult controls. Comparing scores across the different stages of spermatogenesis revealed significantly fewer functionally mature cells in the groups with no or only spermatogonia compared to those with partial differentiation. This was expected since adequate T levels, produced by Leydig cells, are essential for supporting spermatogenesis (*Smith and Walker, 2014*). The regression analysis indicated that age-related differences between the groups could also account for this effect. This could again be attributed to variations in GAHT sensitivity related to age but also to the tendency of Leydig cells to lose functionality with increasing age (*Anand-Ivell et al., 2022*). However, even the oldest control (77 years of age) still had 72% functionally mature Leydig cells. Additionally, this analysis implied that participants with lower FSH levels tended to have fewer functionally mature Leydig cells, aligning with findings by Sapino and colleagues who observed a decrease in Leydig cell numbers correlated with low serum gonadotropin levels (*Sapino et al., 1987*). In this study, functional Leydig cells were identified with the marker CYP11a1 (*Lottrup et al., 2014*). However, as shown by Guo and colleagues, and reflected in the prepubertal controls, CYP11a1 only gradually appears in Leydig cells around puberty (*Guo et al., 2020*), meaning that truly immature Leydig cells could not be identified using this approach. Only the use of an immature Leydig cell marker (delta-like homolog 1) could confirm their presence. Unfortunately, our attempts to optimize the immunofluorescence protocol for this marker were unsuccessful. Due to the negative feedback of estrogens on steroidogenic enzymes, it was considered whether Leydig cells could lose their CYP11a1 signal. While a few cells were CYP11a1$^{-}$/INSL3$^{+}$, most Leydig cells seemed unaffected, suggesting that CYP11a1 expression is persistent to maintain basal T levels (*Table 1*).

Regarding the differential gene expression analysis, this study showed clear segregation between prepubertal and adult control tissue based on spermatogenesis-related genes in P3. Due to the lack of spermatogenesis, transgender testicular tissue clustered with prepubertal rather than adult samples. In fact, it has been shown that the spermatogonia present in transgender tissue express transcripts of early undifferentiated spermatogonia, similar to the early undifferentiated spermatogonia in cisgender tissue of all ages (*Guo et al., 2020*). In addition, also in other expression patterns (P1 and P2), transgender tissue was closer to prepubertal and peripubertal (Peripubertal 1) tissues than to adult tissue. P1 is a pattern that includes biological processes related to tissue morphology and development, and cell location establishment. This is in line with the histological findings regarding the spermatogonial stem cell niche, as well as the findings in the literature (*Guo et al., 2020*) pointing to a rejuvenation of transgender tissue, particularly Sertoli cells. On the other hand, P4 reveals that transgender testicular tissue is deprived of - or at least less rich in - mitotic and meiotic cells in comparison to controls. Also, the observation that seminiferous tubules in transgender tissue shrink (*Leavy et al., 2017*), which may be simply due to germ cell loss, but that they also become hyalinized ghost tubules, indicates that Sertoli cells are not proliferating, but rather depleting. Moreover, we show that transgender Sertoli cells exhibit a phenotype similar to that of Sertoli cells in Sertoli cell-only tubule disorders where immaturity markers and AR are concomitantly expressed (*Sharpe et al., 2003*). Taken together, these findings indicate that Sertoli cells in trans women have only partially dedifferentiated.

Pattern P5 was the most specific for transgender tissue. This pattern, although including a variety of biological processes, points to inflammation and fibrosis, in line with what is observed histologically. Noteworthy, Leydig and PTMCs in the testes of trans women display altered transcriptomes not related to rejuvenation (*Guo et al., 2020*), which may point to potential malignant alterations due to GAHT and to the important role of T in keeping these cells healthy after maturation. Although a recent report shows no increased risk of testicular cancer in GAHT patients (*de Nie et al., 2022b*), several other reports point to a tight correlation between elevated levels of estrogen and germ or Leydig cell cancers (*Huseby, 1980*; *Bouskine et al., 2008*; *Chandhoke et al., 2018*; *Fénichel and Chevalier, 2019*). In another recent report, routine pathology examination following orchidectomy was advised even though malignant findings were rare (*Bonapace-Potvin et al., 2022*). Also interesting to note, the GO term 'response to estrogen' (eight genes, adjusted p-value<1.55e-2) is part of P5, showing

that alterations at the testis level result from a combination of estrogen and anti-androgen therapy, and not solely from T suppression.

A common shortcoming in testis biology and male fertility research, testis toxicology, and endocrine disruption testing is the lack of human testicular tissue, particularly from prepubertal and peripubertal boys. Our findings suggest that transgender testicular tissue which expresses AMH, AR, CYP11a1, INSL3, and ACTA2 contains spermatogonia, and exhibits minimal tubular hyalinization, may serve as a valuable alternative for this. Moreover, a recent study showed that testicular cells from GAHT-treated trans women can preserve their function, and that spermatogenesis can be successfully restored (*de Nie et al., 2023*). The great advantage of using trans women's testicular tissue is that it is abundant since it is treated as medical waste material. This advantage has particular interest for the testing of compounds, whether it be drugs or environmental toxicants, as this application always requires large amounts of samples. In fact, the use of digested trans women's testicular tissue for the testing of compounds in vitro has been described in the literature (*Mincheva et al., 2020*).

While this study provides valuable insights, it is essential to recognize its limitations. The first limitation is that the hormone levels were measured during the last visit before SRS rather than on the day of surgery, and not considering the time of last administration and blood draw, explaining suboptimal E2 serum levels in many participants. Nevertheless, this approach offers insight into the hormonal status throughout treatment, rather than 2 weeks post-GAHT cessation. It is however important to mention that, despite the recommended follow-up interval of 6–12 months (*Hembree et al., 2017*), hormone measurements for four participants dated back more than a year before the surgery. Moreover, for eight participants, only hormone measurements from less than 1 year after the initiation of treatment were available. Another limitation of this study is the lack of information regarding the testicular function of the participants before the start of the hormonal treatment. It is possible that certain transgender-specific factors such as tucking, wearing tight underwear, and having a low masturbation frequency might have negatively impacted their testicular health (*de Nie et al., 2020*). Nevertheless, all trans women in this study showed normal serum T values before initiating GAHT. Finally, it was challenging to draw strong conclusions from the regression analysis within the current cohort, and the available predictors are deemed inadequate for making predictions at an individual participant level. On the other hand, this study features a large cohort of 106 trans women who were subjected to a consistent hormone regimen involving both estrogens and a fixed dose of CPA, ensuring robust statistical analyses and reliable outcomes. Since in ENIGI the practice has changed over the recent years toward reduction in prescribed daily CPA dosage (*Kuijpers et al., 2021*) or abandoning CPA altogether in favor of GnRH analogues as anti-androgen treatment, a re-evaluation in this more recent cohort is advisable.

In summary, our findings show that trans women undergoing GAHT with E2 and CPA exhibit a partially rejuvenated spermatogonial stem cell niche. In the tubular wall, PTMCs continue to express ACTA2 although in a disturbed pattern, likely due to fibrotic processes. Additionally, there is evidence of a regressed seminiferous epithelium and a collapsing lumen. Sertoli cells have partially dedifferentiated (expression of both AMH and AR) and resemble those of peripubertal boys. Leydig cells also show a distribution similar to that of peripubertal tissue and a decreased INSL3 expression. The transcriptomic profiles of transgender and control testicular tissue reveal ongoing inflammation but also confirm the microscopy findings that point to a loss of mature characteristics - a partial rejuvenation - of the spermatogonial stem cell niche in transgender testicular tissue. Hence, testicular tissue from trans women holds the potential to be used as a surrogate for (pre)pubertal tissue in in vitro applications.

## Materials and methods

The collection and use of all tissues were approved by the Committee for Medical Ethics of the Universitair Ziekenhuis (UZ) Brussel - Vrije Universiteit Brussel (VUB) (EC nos. 2016/V9, 2017/061, and 2022/161) and UZ Gent (EC nos. 2009/622, 2014/1175, and 2014/1175-AM01). All patients or their parents gave written informed consent to donate testicular tissue to research.

## Participants

This study is part of the European Network for the Investigation of Gender Incongruence (ENIGI) project, which is a collaborative effort involving prominent gender identity clinics in Western Europe. The current study population consists of the one (n=97) used in our previous publication (*Vereecke et al., 2020*), plus nine other participants, totaling n=106 trans women exclusively selected from the UZ Gent.

## Gender-affirming hormone therapy

The included trans women underwent a standardized diagnostic procedure and were at least 16 years of age before starting GAHT (*Dekker et al., 2016*). Treatment protocols were in accordance with the World Professional Association for Transgender Health Standards of Care, Version 7 (WPATH SOC 7) (*Coleman et al., 2022*), and consisted of an oral administration of 2 mg of E2 valerate (ProgynovaVR, Bayer, Diegem, Belgium) twice daily, along with 50 mg of the anti-androgen CPA (AndrocurVR, Bayer, Diegem, Belgium) once daily. For participants aged 45 and above and participants with a previous history of venous thromboembolism, the estrogen treatment consisted of transdermal E2 (Derme-strilVR, Besins, Brussels, Belgium) at a daily dosage of 50–100 mg. Additionally, participants who experienced intolerance to the initial estrogen treatment were given 2 mg of transdermal 17-b E2 gel (OestrogelVR, Besins, Brussels, Belgium) twice daily (*Dekker et al., 2016*). To reduce the risk of peri-operative thromboembolism, hormone therapy was stopped 2 weeks before SRS.

## Laboratory analysis

During the last pre-operative visit, a blood sample was obtained to measure the participant's reproductive hormone levels (LH, FSH, T, and E2) and to assess the Sertoli cell function (AMH and inhibin B). Competitive chemiluminescent immunoassays were performed for the quantification of LH (E170 Modular, Roche, Gen III, interassay coefficient of variation [CV] 3.48%, limit of quantification [LOQ] 0.1 mIU/mL), FSH (E170 Modular, Roche, Gen III, interassay CV 3.3%, LOQ 0.1 mIU/mL), T (E170 Modular, Roche, Gen II, LOQ 10 ng/dL) [0.4 nmol/L], interassay CV 2.6%, and E2 (E170 Modular, Roche, Gen III, with an LOQ of 25 pg/mL and an interassay CV of 3.2%). Prior to March 19, 2015, E2 levels were assessed using an E170 Modular (Gen II, Roche Diagnostics, Mannheim, Germany). To convert these E2 values, the following formula was employed: Gen III = 6.687940 + 0.834495 * Gen II. AMH was assessed using an electrochemiluminescence immunoassay (Cobas E801, Roche Diagnostics, Mannheim, Germany), with an interassay CV of 4.4% and an LOQ of 0.01 mg/L. Inhibin B was quantified through an enzyme-linked immunosorbent assay (Beckman Coulter, Immunotech, Prague, Czech Republic), which exhibited an interassay CV of 6.6% and had an LOQ of 2.91 pg/mL. To score the hormone levels, the reference values from the device package inserts were utilized, unless explicitly stated otherwise (*Supplementary file 1*; *Supplementary file 2*; *Supplementary file 3*; *Supplementary file 4*; *Supplementary file 5*; *Supplementary file 6*).

## Tissue sampling

As part of their SRS, all transgender participants underwent a bilateral orchidectomy at the UZ Gent. For the 97 participants included in our previous publication, testicular tissues were treated as described (*Vereecke et al., 2020*). For the nine more recent participants, the tunica albuginea and, if applicable, the rete testis were removed and the testes were processed in pieces of approximately 50 mm$^3$. Tissue pieces were collected in Dulbecco's modified essential medium/F12 (DMEM/F12, 11320-033, Gibco, New York, USA) and were pseudomized by UZ Gent and transported to the VUB on ice. At VUB, tissue pieces were either fixed in AFA for 1 hr, transferred to formalin, dehydrated and, subsequently, paraffin-embedded, or were sectioned into small fragments and cryopreserved by slow freezing (*Baert et al., 2013*). Briefly, regarding cryopreservation, tissue pieces (1–6 mm$^3$) were equilibrated for 10 min on ice in cryomedium consisting of 1.5 M dimethylsulfoxide (DMSO, D2650, Sigma-Aldrich, Hoeilaart, Belgium), 0.15 M sucrose (D21881000, PanReac AppliChem, Darmstadt, Germany) and 10 mg/mL (10%) human serum albumin (HSA, 10064, Vitrolife, Londerzeel, Belgium) diluted in DMEM/F12 (11320-033, Gibco, New York, USA). Before being stored in liquid nitrogen, the cryovials were placed in an isopropyl alcohol container (Mr. Frosty, VWR, PA, USA) for 24 hr at –80°C.

## Tissue for stainings

Transgender testicular tissue embedded in paraffin was sectioned at 5 µm. Control adult testicular tissue was acquired from six men who underwent orchidectomy or vasectomy reversal in the UZ

**Table 2.** Testicular tissue characterization for gene expression analysis.

| Sample name | Age at biopsy/ orchidectomy (years) | Histological notes |
| --- | --- | --- |
| Prepubertal 1 | 1 | Spermatogonia |
| Prepubertal 2 | 4 | Spermatogonia |
| Prepubertal 3 | 1 | Spermatogonia |
| Peripubertal 1 | 13 | Spermatocytes<br>Absent seminiferous lumen |
| Peripubertal 2 | 15 | Spermatocytes<br>Half-open seminiferous lumen |
| Peripubertal 3 | 12 | Elongated spermatids<br>Half-open seminiferous lumen |
| Adult 1 | 72 | Normal spermatogenesis<br>Orchidectomy |
| Adult 2 | 84 | Normal spermatogenesis<br>Orchidectomy |
| Adult 3 | 38 | Normal spermatogenesis<br>Re-anastomosis |
| Adult 4 | 73 | Normal spermatogenesis<br>Orchidectomy |
| Adult 5 | 78 | Normal spermatogenesis, signs of fibrosis orchidectomy |
| Trans woman 1 | 18 | Spermatogonia |
| Trans woman 2 | 19 | Spermatogonia |
| Trans woman 3 | 25 | Spermatogonia |
| Trans woman 4 | 26 | Spermatogonia |
| Trans woman 5 | 34 | Spermatogonia |
| Trans woman 6 | 44 | Sertoli cell only |

Brussel. Additionally, peripubertal (n=3) and prepubertal (n=3) testicular tissues were obtained from patients who underwent a testicular biopsy in the UZ Brussel in the context of fertility preservation. These control tissues were selected based on the patients' age at banking and the stage of germ cell differentiation. The peripubertal controls ranged from 12.0 to 13.4 years of age, demonstrated ongoing spermatogenesis, and thus represented (early) puberty. The prepubertal controls, aged 3.5–5.5 years, did not exhibit ongoing spermatogenesis. All patients or their parents gave written informed consent to donate a piece of testicular tissue to research. After removal, the testicular biopsies were transported to the BITE laboratory on ice, where they were washed and fixated in acidified alcoholic formalin (AFA0020AF59001, VWR, Leuven, Belgium) for 1 hr. Subsequently, the fixed samples were sent to the pathology department of the UZ Brussel for overnight fixation in formalin and embedding in paraffin. All control tissues were sectioned at 5 µm in the BITE Laboratory.

### Tissue for gene expression
For gene expression analysis, cryopreserved testicular tissue from six trans women, and previously cryopreserved tissue from five adult, three peripubertal and three prepubertal controls were used (*Table 2*).

### Stainings
*Table 3* provides an overview of the used markers.

### General histology and germ cell differentiation
The overall histology of the samples was evaluated by a hematoxylin-eosin or hematoxylin-periodic acid Schiff (PAS) staining in accordance with the manufacturer's protocol. The degree of germ cell

**Table 3.** Antibody specifications.

| Antibody | Target cells | Dilution | Reference number | Company |
|---|---|---|---|---|
| MAGE-A4 | Spermatogonia and primary spermatocytes | 1/200 | / | Provided by Dr Giulio Spagnoli, University of Basel, Switzerland |
| BOLL | Secondary spermatocytes and round spermatids | 1/400 | H0006637 | Novusbio Bio-Techne, Oxon, UK |
| CREM | Round spermatids | 1/2000 | hpa001818 | Merck, Overijse, Belgium |
| ACROSIN | Round, elongating and elongated spermatids | 1/500 | sc67151 | Tebu-bio, Boechout, Belgium |
| ACTA2 | Peritubular myoid cells | 1/2000 | A2547 | Sigma-Aldrich, Overijse, Belgium |
| AMH | Immature Sertoli cells | 1/200 | MCA2246 | Bio-Rad, Temse, Belgium |
| AR | Mature Sertoli cells | 1/2000 | ab133273 | Abcam, Cambridge, UK |
| SOX9 | Sertoli cells | 1/200 | AB5535 | EMD Millipores, Overijse, Belgium |
| INSL3 | Mature Leydig cells | 1/2000 | HPA028615 | Sigma-Aldrich, Machelen, Belgium |
| CYP11a1 | Functional Leydig cells | 1/50 | 13363-I-AP | Proteintech, Manchester, UK |
| Goat anti-mouse-HRP | / | 1/200 | P0447 | Agilent Technologies, Glostrup, Denmark |
| Goat anti-rabbit-HRP | / | 1/200 | PI-1000 | Vector Laboratories, CA, USA |
| Goat anti-rabbit Alexa Fluor 647 | / | 1/200 | A21245 | Life Technologies, Carlsbad, CA, USA |

differentiation was determined through staining for melanoma-associated antigen A4 (MAGE-A4), boule homolog RNA-binding protein (BOLL), cAMP-responsive element modulator (CREM), and ACROSIN. These markers identify spermatogonia/primary spermatocytes, secondary spermatocytes/round spermatids, round spermatids, and round/elongating/elongated spermatids, respectively (*Peri and Serio, 2000*; *Muciaccia et al., 2013*; *de Michele et al., 2018*; *Guo et al., 2018*). Notably, germ cell staining results from 97 participants have already been reported (*Vereecke et al., 2020*) and were integrated into the current paper's findings.

## Peritubular myoid cells

The status of the peritubular myoid cells was determined by the presence of ACTA2 (*Schlatt et al., 1993*). Deparaffinized and rehydrated sections were treated with an $H_2O_2$/methanol solution for 30 min to eliminate endogenous peroxidase activity. Next, antigen retrieval was conducted in citric acid (made in-house, pH 6), using a water bath at 98°C for 75 min. Then, the sections were incubated with 10% normal goat serum (NGS, B304, Tebu-bio, Boechout, Belgium)/1% bovine serum albumin (BSA, 10735094001, Roche Diagnostics, Vilvoorde, Belgium) in phosphate-buffered saline (PBS, 70011-036, Life Technologies, Carlsbad, CA, USA) for 30 min. After this, all sections, except for the negative control, were exposed to the primary antibody ACTA2. The sections were then incubated overnight in a humidified chamber at 4°C. On the following day, the slides were treated with the Dako Real Envision Detection System kit (K500711-2, Agilent, Santa Clara, CA, USA) and counterstained with hematoxylin.

## Sertoli cells

Sertoli cell maturation was assessed by the general marker SOX9, maturity marker AR, and immaturity marker AMH (*Guo et al., 2020*). After deparaffinization and rehydration, antigen retrieval was

performed in a Tris-ethylenediamine tetra-acetic acid (EDTA) buffer (homemade, pH 9) using a water bath at 95°C for 75 min. Next, all sections were incubated with an $H_2O_2$/methanol solution for 30 min. After blocking the slides with 4% NGS in PBS for 1 hr, they were incubated with the primary antibody AMH overnight. The sections were exposed to the secondary antibody goat anti-mouse-horseradish peroxidase (HRP) for an hour, followed by fluorescein (NEL741001KT, Akoya Biosciences, Marlborough, MA, USA) for 5 min to develop the first color. Subsequently, the sections underwent a second antigen retrieval in a microwave for two cycles of 5 min at 500 W, in Tris-EDTA. After this, the sections were blocked with 20% NGS/5% BSA in Tris-buffered saline (TBS, homemade, pH 7.4) for 30 min and incubated overnight with the primary antibody AR. The following day, the secondary antibody goat anti-rabbit-HRP was added for 30 min, after which the second color was generated by Cyanine 3 incubation (NEL744001KT, Akoya Biosciences, Marlborough, MA, USA) for 5 min. The final antigen retrieval was achieved in a 75 min water bath at 95°C in Tris-EDTA. The sections were then blocked with 5% NGS in TBS for 1 hr. Subsequently, the third primary antibody SOX9 was incubated overnight. The last secondary antibody, goat-anti-rabbit Alexa Fluor 647, was incubated for 1 hr, after which the slides were exposed to Hoechst (H3570, Life Technologies, Carlsbad, CA, USA, 1/2000) for 10 min. Finally, the slides were mounted using Prolong Gold antifade reagent (P36934, Invitrogen, Thermo Fisher Scientific, Breda, The Netherlands) and stored in the dark at 4°C.

## Leydig cells

The maturity marker INSL3 and functional marker CYP11a1 were used to assess the status of the Leydig cells (*Lottrup et al., 2014*). Following deparaffinization and rehydration, antigen retrieval was performed in the microwave for two cycles of 5 min at 500 W in Tris-EDTA. Subsequently, all sections were incubated with 0.1% Tween-20 (P1379-250ML, Sigma-Aldrich, Overijse, Belgium)/3% $H_2O_2$ in TBS, followed by 20% NGS/5% BSA in TBS for 30 min each. After this, the sections were incubated overnight with the primary antibody INSL3. The secondary antibody goat anti-rabbit-HRP was added for 30 min, after which fluorescein was developed for 5 min. The second antigen retrieval took place in a 75 min water bath at 95°C in Tris-EDTA. After another blocking step with 5% NGS in PBS for an hour, the sections were exposed to the primary antibody CYP11a1 overnight. The next day, the secondary antibody goat-anti-rabbit Alexa Fluor 647 was added for an hour, and Hoechst was added for 10 min. The sections were mounted with ProLong Gold antifade reagent and stored in the dark at 4°C.

## Microscopic analysis

Histological examination was performed on an inverted microscope (IX81, Olympus, Aartselaar, Belgium). Four representative images within one depth were captured for each participant using the imaging software CellF (version 2.8, Olympus, Aartselaar, Belgium) at a magnification of ×10. Damage to the testicular niche was evaluated on PAS stainings by three independent individuals and involved scoring of the tubular lumen and tubular hyalinization. One overall score was addressed for each participant using modified classification scales from *de Nie et al., 2022b*. The lumen was defined as 'open', 'half-open', or 'absent' (*Figure 1A*). Tubular hyalinization was scored based on the presence and thickness of a hyaline region separating the peritubular layer from the basal membrane of the seminiferous tubule (*Figure 1B*). The absence of hyalinization was scored as 'no', while subtle hyalinization was categorized as 'mild'. When a distinct, thick hyaline region could be observed, the tubule was scored as 'moderate', and when the tubule was nearly entirely hyalinized, it was classified as 'severe'. To determine the stage of germ cell differentiation, images from the MAGE-A4, BOLL, CREM, and ACROSIN stainings were scored as positive or negative for each respective marker (*Vereecke et al., 2020*; *Figure 1C*). To evaluate the PTMCs, the ACTA2 staining patterns in trans women were compared to those observed in the controls. As shown in *Figure 2*, four ACTA2 patterns could be established: 'intact' (adult-like), 'disconnected', 'interrupted' (peripubertal-like), and 'absent' (prepubertal-like).

The immunofluorescent stainings were scanned with an axioscanner (Axioscan 7, Zeiss, Zaventem, Belgium). Four representative images within one depth were captured for each sample using the ZEN software (version 3.5, Zeiss, Zaventem, Belgium). For the Sertoli cell stainings, each image covered 2000 μm, whereas for the Leydig cell pictures, the images covered 1000 μm each. The images were subsequently analyzed with ImageJ (version 2.9.0/1.53t, National Institute of Health, Bethesda, MD, USA). All seminiferous tubules within the four images were assessed for the presence or absence

of Sertoli cell markers SOX9 (general marker), AR (maturity marker), and AMH (immaturity marker) (*Figure 3*). For each participant, the outcomes were presented as the percentage of 'mature' (SOX9$^+$/AR$^+$/AMH$^-$), 'semi-mature' (SOX9$^+$/AR$^+$/AMH$^+$), and 'immature' (SOX9$^+$/AR$^-$/AMH$^-$) tubules. Besides this, the CYP11a1-expressing Leydig cells (functional marker) within the four images were quantified and scored as either positive or negative for INSL3 (maturity marker) (*Figure 4*). For each individual, results were expressed as the percentage of 'functionally mature' (CYP11a1$^+$/INSL3$^+$) Leydig cells.

## Statistical analysis

The statistical analysis was performed using R version 4.3.2 (R Core Team, Vienna, Austria). Data are displayed as the median (range) or as the mean ± SD. For each of the different research questions, different statistical choices were made, as will be explained in the Results section. It is worth mentioning here that the distributions of the data of interest present some challenges for their analysis, and different techniques are used to deal with those challenges. The data for the Sertoli and Leydig cells have many observations at the extremes of the scale (0% or 100%) and the different percentages are interdependent, for which a hyperbolic tangent transformation is used on specific ratios of these percentages. Certain predictors are also transformed or altered, because of too strong intercorrelation and skewness. A new variable was created that combined FSH and LH, creating three categories, those with FSH under 0.3, and those with FSH over 0.3 but either LH under 0.1 or over or equal to 0.1. A log-transformation of the E2 and T predictors was also considered. As these two are also strongly correlated they are never combined within one model.

## RNA extraction and RNA sequencing

Cryopreserved testicular tissue was thawed and cryoprotectants were osmotically removed according to *Baert et al., 2013*. RNA extraction was performed with the QIAGEN RNeasy Micro Kit (QIAGEN, Hilden, Germany). RNA concentration after extraction was assessed by using the NanoDrop ND-1000 UV-Vis Spectrophotometer (Thermo Fisher Scientific, Breda, The Netherlands). Using the DNF-472 High Sensitivity RNA Analysis Kit, the quality of the RNA samples was assessed on the AATI Fragment Analyzer (Agilent Technologies Inc, Santa Clara, CA, USA). RNA libraries were created from 150 ng of total RNA using the KAPA RNA HyperPrep Kit with RiboErase kit (Roche Diagnostics, Vilvoorde, Belgium), according to the manufacturer's directions. In summary, following ribodepletion and DNase digestion, RNA was fragmented to average sizes of 200–300 bp by incubating the samples for 6 min at 94°C. After first-strand synthesis, second-strand synthesis and adapter ligation, the libraries were amplified using 12 PCR cycles. Using the DNF-474 High Sensitivity NGS Fragment Analysis Kit, final libraries were qualified on the AATI Fragment Analyzer (Agilent Technologies Inc, Santa Clara, CA, USA), and quantified on the Qubit 2.0 with the Qubit dsDNA HS Assay Kit (Life Technologies, Carlsbad, CA, USA). Using the NovaSeq 6000 S4 Reagent Kit (200 cycles), 25 million 2×100 bp reads were generated per sample on the Illumina NovaSeq 6000 system (Illumina Inc, San Diego, CA, USA). For this, 1.9 nM libraries were denatured according to the manufacturer's directions. After demultiplexing and an adaptor/quality trimming step, the raw reads were mapped against the human genome (hg19) using Spliced Transcripts Alignment to a Reference - STAR (*Dobin et al., 2013*), and then translated into a quantitative measure of gene expression with the tool HTSeq (*Anders et al., 2015*).

## Differential gene expression and functional analysis

The UMI matrix was normalized with the regularized log (rlog) transformation package from DeSeq2 (*Love et al., 2014*). The AMEN suite was used to identify DEGs (*Chalmel and Primig, 2008*). A statistical comparison between the different types of testicular tissue was made to identify DEGs. In short, genes that were more highly expressed than the background cutoff (overall median of rlog-transformed UMI dataset, 2.03), and that exceeded 2.0-fold change compared to the control, were used for further analysis. Significant DEGs were identified by using the empirical Bayes moderated t-statistics implemented into the LIMMA package with an adjusted F-value estimated using Benjamini and Hochberg false discovery rate (FDR) approach ($p \leq 0.05$) (*Smyth, 2004*; *Ritchie et al., 2015*). Partitioning of the DEGs was performed by using the k-means method. Expression profiles of DEGs were displayed as false-color heatmaps using the 'pheatmap' package. Functional analyses were performed with AMEN (*Chalmel and Primig, 2008*) with an FDR-adjusted p-value of $\leq 0.05$.

## Acknowledgements

The authors would like to express their gratitude to Dr. Margo Willems and M.Sc. Julie Kerckx for their pivotal role in optimizing the staining protocols, and to Dr. Justine Defreyne, Dr. Sarah Collet, and MD Jeroen Vervalcke for their invaluable assistance in collecting and managing the participant data. To Sylvie Lierman for processing the donated testis samples at UZ Gent. Additionally, the authors deeply appreciate the essential support, provision of facilities, and resources from the Endocrinology and Pathology department of the UZ Gent (with special recognition for Prof. Jo Van Dorpe), and the Pathology department of the UZ Brussel. Finally, our gratitude extends to all participants involved in the ENIGI study protocol.

## Additional information

### Funding

| Funder | Grant reference number | Author |
| --- | --- | --- |
| Scientific fund Willy Gepts | | Yoni Baert |
| Fonds Wetenschappelijk Onderzoek | G026223N | Yoni Baert |
| Strategic research program 89 Vrije Universiteit Brussel | SRP89 | Ellen Goossens |

The funders had no role in study design, data collection and interpretation, or the decision to submit the work for publication.

### Author contributions

Emily Delgouffe, Data curation, Formal analysis, Investigation, Visualization, Methodology, Writing – original draft, Project administration, Writing – review and editing, She conducted the staining procedures, performed the imaging and analysis, and processed the data; Samuel Madureira Silva, Data curation, Formal analysis, Investigation, Visualization, Methodology, Writing – original draft, Writing – review and editing, He gathered the tissue samples for gene expression analysis, performed the RNA extraction, and the final gene expression analysis, and drafted the part on gene expression analysis; Frédéric Chalmel, Resources, Formal analysis, Validation, Investigation, Methodology, He performed the differential gene expression and functional analysis; Wilfried Cools, Data curation, He conducted the statistical analysis; Camille Raets, Data curation, She conducted the statistical analysis; Kelly Tilleman, Project administration, Writing – review and editing, She managed the recruitment and consent of the trans women for the gene expression analysis; Guy T'Sjoen, Resources, Project administration, Writing – review and editing, He managed the recruitment of the trans women and their clinical data; Yoni Baert, Conceptualization, Supervision, Funding acquisition, Validation, Methodology, Writing – review and editing; Ellen Goossens, Conceptualization, Resources, Supervision, Funding acquisition, Validation, Writing – review and editing

### Author ORCIDs

Emily Delgouffe [ID] https://orcid.org/0000-0001-5611-2173
Samuel Madureira Silva [ID] http://orcid.org/0000-0002-0833-123X

### Ethics

Human subjects: The collection and use of all tissues were approved by the Committee for Medical Ethics of the Universitair Ziekenhuis (UZ) Brussel - Vrije Universiteit Brussel (VUB) (EC nos. 2016/V9, 2017/061 and 2022/161) and UZ Gent (EC nos. 2009/622, 2014/1175 and 2014/1175-AM01). All patients or their parents gave written informed consent to donate testicular tissue to research.

Reviewer #1 (Public Review): https://doi.org/10.7554/eLife.94825.3.sa1
Reviewer #2 (Public Review): https://doi.org/10.7554/eLife.94825.3.sa2
Author response https://doi.org/10.7554/eLife.94825.3.sa3

# Additional files

## Supplementary files

Supplementary file 1. Reference values for luteinizing hormone.

Supplementary file 2. Reference values for follicle-stimulating hormone.

Supplementary file 3. Reference values for testosterone.

Supplementary file 4. Reference values for estradiol.

Supplementary file 5. Reference values for anti-Müllerian hormone.

Supplementary file 6. Reference values for inhibin B.

Supplementary file 7. Full list of GO terms enrichment analysis.

MDAR checklist

## Data availability

The datasets supporting the findings of this study are stored under restricted access in the VUB Institutional Data Repository, under the accession numbers VUB/BITE/1/000003 and VUB/GRAD/1/000001, due to participant privacy concerns. The first dataset contains the clinical data and histological characterisation of the patients, while the latter includes the RNA-seq raw and preprocessed data. Access to the data will be considered on a case-by-case basis and must be requested by contacting Prof. Ellen Goossens (ellen.goossens@vub.be), who will review the request, including the intended purpose of the research and any potential commercial use. A data use agreement must be completed and signed in accordance with the VUB legal department's guidelines before any data can be shared or released. Metadata for the datasets can be accessed through the VUB Research Portal at: https://researchportal.vub.be/en/datasets/characterization-of-testicular-tissue-in-trans-women and https://researchportal.vub.be/en/datasets/partial-rejuvenation-of-the-spermatogonial-stem-cell-niche-after-.

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
