## [Editor Report · eLife assessment]

This **important** study presents new knowledge of the spermatogonial stem cell (SSC) niche in trans women after gender-affirming hormone therapy (GAHT). The evidence supporting the claims is **convincing**. The work will be of interest to researchers and clinicians working in the field of reproductive medicine and andrology.

---

## [Referee Report · Reviewer #1 (Public Review)]

Summary:

This is a nice paper taking a broad range of aspects and endpoints into account. The effect of GAHT in girls has been nicely worked out. Changes in Sertoli and peritubular cells appear valid, less strong evidence is provided for Leydig cell development. The recovery of SSCs appears an overjudgement and should be rephrased. The multitude and diversity of datasets appear a strength and a weakness as some datasets were not sufficiently critically reviewed and a selection of highlights provides a certain bias to the interpretation and conclusion of the study.

The authors need to indicate that the subset of data on SSCs has been reported previously Human Reprod 36: 5-15 (2021) and is simply re-incorporated in the present paper. as Fig. 1C. There are sufficient new results to publish the remaining datasets as a separate paper. Authors could refer to the SSC data with reference to the previous publication.

Strengths:

The patient cohort is impressive and is nicely characterized. Here, histological endpoints and endocrine profiles were analyzed appropriately for most endpoints. The paper is well-written and has many new findings.

Weaknesses:

The patients and controls are poorly separated in regard to pubertal status. Here additional endpoints (e.g. Tanner status) would have been helpful especially as the individual patient history is unknown. Pre- and peri-puberty is a very rough differentiation. The characterization and evaluation of Leydig cells is the weakest histological endpoint. Here, additional markers may be required. Fig. 1 suffers from suboptimal micrograph quality.

---

## [Referee Report · Reviewer #2 (Public Review)]

Summary:

The study is devoted to the deep investigation of the spermatogonial stem cell (SSC) niche in trans women after gender-affirming hormone therapy (GAHT). Both cellular structure and functionality of the niche were studied. The authors evidently demonstrated that all cellular components of SSC niche were affected by hormone therapy. Interestingly, the signs of "rejuvenation" within the niche were also observed indicating the possible reverse to the immature condition.

Strengths:

The obtained findings are important for the better understanding of hormonal regulation of testis and SSC niche and provide some clues for using the biomaterials from these specific and even unique donors for biomedical research.

Weaknesses:

This study has some limitations. Many studies can't be done using the testes cells of trans women, since their cells are significantly different from adult man cells and less from prepubertal and pubertal cells. The authors themselves identify some of the limitations: this material is suitable only for studying prepubertal processes in the testis. However, the authors also report large variability in data due to different hormonal therapy regimens and, apparently, age. Accordingly, not all material obtained from trans women can also be used for studies of prepubertal processes.

---

## [Author Response]

The following is the authors’ response to the original reviews.

**Reviewer #1 (Recommendations For The Authors):**
(1) Data on SSCs are published from a previous report (Fig. 1C). These should be deleted or marked as such.

We acknowledge the need for clarification regarding our study population for the germ cell stainings. As stated in our Materials and Methods section, our current study population includes the cohort from our previous publication (Vereecke et al., 2020), supplemented by nine additional participants, totaling n=106 trans women. Fig. 1C incorporates both previous and new data on germ cells, and this was further clarified in the Materials and Methods section.

(2) Many micrographs are suboptimal and need to be replaced by better photos presenting cellular details more clearly.

The Figures were remade to solve the suboptimal resolution.

(3) Table 2 would benefit from a column indicating the target cell or organelle.

This column was added to Table 2.

(4) The pubertal status is poorly defined by pre- and peripubertal terms. The authors should add more informative clinical scores.

We included information on the Tanner stages of the trans women in our cohort (all G5), as well as details on the selection criteria for our controls and their pubertal status.

(5) The characterization of Leydig cells is incomplete. Several better markers would validate the findings.

As briefly touched upon in the discussion, the marker delta-like homolog 1 would indeed be valuable to assess the presence of truly immature Leydig cells. Unfortunately, our attempts to optimize the immunofluorescence protocol for this marker were unsuccessful, resulting in a double staining instead of a triple staining for the Leydig cells. This statement was also added to the Discussion.

(6) The selection bias for datasets is obvious. It seems that the authors try to create nice stories but do not always refer to less compelling datasets. Here a more critical view may be necessary to gain a more realistic view and may open alternative explanations.

We would appreciate clarification on which datasets may have been insufficiently reviewed and how our selection of highlights may have introduced bias to the interpretation and conclusion of the study. It is important to note that we did not select any patients/ data; all patient data were incorporated into our results section.

(7) The term rejuvenation for the stem cell niche/germ cell complement is misleading in the title and text. Could the authors consider another team e.g... restoration., (de)differentiation. Alternatively, define the term juvenation in a more substantial manner.

We did not change the term “partial rejuvenation” as we believe it best describes our findings. We did however introduce the term in a more substantial manner in our Abstract and Discussion.

**Reviewer #2 (Recommendations For The Authors):**
(1) The authors provided a lot of scattered data, but it would be useful to formulate clear criteria (hormonal therapy, age, end points, etc.) that the material must meet so that it can be used for research into prepubertal processes.

We have added these criteria to our Discussion. However, our current results do not yet reveal how these tissues behave in vitro. Ongoing research is addressing this question and will be presented in a future paper.

(2) Is there any research on the preservation of functions of testicular cells from trans women?

This data would be very useful, for example, for models for drug testing. Yes: a reference to this paper was added to our Discussion.

(3) It is recommended to present the data in a table reflecting the correlations found by the authors and the correlations from the literature between cellular changes and hormone levels and age.

After careful consideration, we have decided to proceed without incorporating these suggested changes. Our paper focuses on original findings rather than synthesizing existing literature. As such, we have chosen to emphasize our novel results and to compare them to the existing literature in the discussion section.

(4) The authors can also provide data on clinical standards for hormone levels depending on gender and age.

This was added as Supplementary Tables 1-6.

(5) It is recommended to add links to sources from which information about cellular prepubertal, pubertal and adult markers was taken.

This information was added throughout the manuscript.

(6) Is it known which cells within the wall of the seminiferous tubules in adults express AMH? Please clarify.

It has been shown that AMH receptor type 2 starts to be expressed in peritubular mesenchymal cells within the tubular walls during puberty and it remains so throughout adulthood (Sansone et al., 2020). AMH bound to this receptor may help explain the observed AMH signal in the tubular wall of peripubertal and adult controls. This information was added to our Discussion.

(7) How was the degree of hyalinization assessed? It's not obvious from the pictures.

This was further clarified in the Materials & Methods section.

(8) Why were inhibin B and AMH not measured in all patients?

Inhibin B and AMH levels were not available for all patients due to the retrospective nature of these analyses. The measurements were not consistently recorded for all individuals within the historical dataset upon which our research relies.

(9) Why does picture 3A present few SOX9 on adult Sertoli cells, although this is their typical marker?

SOX9 was present in the adult Sertoli cells. However, this signal appears to be more "diluted" in adults due to their ongoing spermatogenesis.